# Identification and Evolutionary Relationship of *Corynebacterium striatum* Clinical Isolates

**DOI:** 10.3390/pathogens11091012

**Published:** 2022-09-05

**Authors:** Jiao Wang, Jiao Pei, Mingming Liu, Rui Huang, Jiqiang Li, Shiying Liao, Jian Liang

**Affiliations:** 1School of Basic Medicine, Hubei University of Arts and Science, Xiangyang 441053, China; 2Xiangyang Central Hospital, Affiliated Hospital of Hubei University of Arts and Science, Xiangyang 441021, China

**Keywords:** *Corynebacterium striatum*, multi-drug resistance (MDR), MALDI-TOF MS, Phylogeny, relationship between microbiota evolution and drug resistance

## Abstract

*Corynebacterium striatum* has developed into a new community-acquired and hospital-acquired multi-drug resistance (MDR) bacterium, and is a potential target pathogen for infection control and antibacterial management projects. In this study, non-duplicate samples of inpatients were collected from a local central hospital. Mass spectrometry showed that 54 *C. striatum* isolates mainly appeared in secretion and sputum from 14 departments. Protein fingerprint cluster analysis showed that the isolates were divided into four groups, most of which appeared in summer. The drug resistance test showed that all strains had multi-drug resistance, with high resistance rates to lincosamides, quinolones and tetracycline detected. Further analysis of the phylogenetic tree of *C. striatum* was conducted by cloning the *16S rRNA* gene. It was found that isolates in the same department had high homology and tended to be located in the same branch or to be crossed in the same main branch. The strains in the same evolutionary branch group had the same drug resistance. Screening of site-specific recombinant elements revealed that 18 strains had *integrase* genes with the same sequence. This study shows that there may be mobile genetic elements in clinical isolates that drive gene exchange among strains, thus causing the cross-infection, spread and evolution of pathogenic bacteria in the hospital.

## 1. Introduction

*Corynebacterium striatum* (*C. striatum*) is a Gram-positive bacterium of the genus *Corynebacterium*. It has normally been reported to parasitize human skin, nasopharynx and other mucosal sites, but in the last decade it has increasingly been reported to be a community-acquired and nosocomial infectious opportunistic pathogen. This bacterium can form biofilms, enhance its survival rate in host and hospital environments, and increase its tolerance to antibiotics and the immune system [1]. It causes bacteremia, sepsis, septic arthritis, meningitis, synovitis, sinusitis, osteomyelitis and other diseases of different degrees in patients with chronic diseases and low immunity [2,3]. It has also been found to be a zoonotic pathogen [4]. In the past, *C. striatum* were often misjudged as contaminating bacteria and ignored in clinical identification. Today, the multi-drug resistance (MDR) of *C. striatum*, as an emerging pathogen, has become one of the public health problems of global concern due to the expression of the MDR spectrum and the study of virulence mechanisms [5,6].

Recently, the clinical detection rate of *C. striatum* has been increasing year by year. Many scholars have found, in exploring the drug resistance mechanism of this bacteria, that its chromosomes encode multiple drug resistance genes, such as the ribosome methylase gene related to macrolide antibiotics (*ermX*), *tetA/B* for tetracycline, oxytetracycline and oxacillin, and *cmx* and *aphA1* for aminoglycosides and chloramphenicol [7]. The ribosomal protection protein gene *tetW* associated with tetracycline resistance can be transmitted in parallel between different strains through plasmids [8], resulting in the strains developing antibiotic resistance. Double mutations in the *gryA* gene show resistance to fluoroquinolones [9]; *AAC(3)-XI* encodes an aminoglycoside (3-N-acetyltransferase) and thus is resistant to aminoglycoside antibiotics [10]. Currently, vancomycin, teicoplanin and linezolid are the most effective drugs against *Corynebacterium*, but clinical reports of vancomycin resistance [11] have made the development of new medicines with bactericidal properties more urgent [12]. More worryingly, the recent global outbreak of 2019-nCoV will further exacerbate global antibiotic resistance [13,14].

The whole genome sequencing of *C. striatum* has facilitated the investigation of its genomic characteristics, understanding of pathogen molecular epidemiology, and global spread and virulence mechanism research [5]. There are two drug resistance mechanisms, endogenous and exogenous: the former usually involves chromosomally encoded elements that acquire phenotypes by regulating membrane permeability and nonspecific efflux pumps [15]. The latter drives the spread and evolution of MDR *C. striatum* through horizontal gene transfer, that is, through mobile genetic elements (MGEs), such as plasmids, transposons, phages, integrons, etc., that carry resistance genes [16,17,18,19]. To better understand the development and evolutionary diversity of *C. striatum* resistance, this study collected a total of 54 MDR *C. striatum* clinical isolates from 2019 to 2020 using mass spectrometry and *16S rRNA* gene identification. Phylogenetic analysis showed that the homology between the flora in the same ward patients was extremely high, the drug resistance was similar, and there was a vital component of the mobile element, the *integrase* gene [20,21], suggesting that *C. striatum*’s pathogenic bacteria resistance is dynamic and expanding. This research on the mechanism of drug resistance transmission and the evolution of opportunistic pathogens should be implemented promptly.

## 2. Results

### 2.1. Clinical Characteristics of the Isolated Specimens

Cream-colored, raised colonies with a diameter of about 1 mm were isolated and purified on Columbia blood agar plates, and 54 strains of *C. striatum* were identified by IVD MALDI Biotyper mass spectrometer as Cs1–Cs54. These strains were isolated from different types of clinical specimens. Secretion specimens (secretion of skin and soft tissues) were the most common (32), followed by sputum (15), urine (3) and blood (4) specimens (Table 1). The strains were isolated from a wide distribution of departments, mainly orthopedics (22) and neurology (8), followed by nephrology (6) and critical care medicine (5). In traditional Chinese medicine, emergency ICU, endocrinology, obstetrics, head and neck oncology, gynecology, otorhinolaryngology head and neck surgery, county hospital, respiratory medicine, and neurosurgery, there were 1–3 cases of infection. *C. striatum* was mainly isolated from secretion-type specimens in orthopedics, while sputum was the primary type in neurology and critical care medicine departments (Table 1). Further analysis of the clinical diseases of the corresponding patients found three cases of diabetes mellitus, six cases of cancer and cysts, two cases of gastrointestinal bleeding, one case of bacteremia, two cases of pneumonia, four cases of chronic kidney disease, one case of systemic lupus erythematosus, nine cases of paralysis, pressure ulcers and pressure ulcers, three cases of cerebral infarction, one case of left lower extremity arterial embolism, one case of fracture, three cases of skin and soft tissue injury, one case of tissue ulcer, one case of chronic infection, two cases of craniocerebral damage, four cases of disturbance of consciousness, and one case of CO poisoning (Appendix A). Among 54 patients, 37 were male, and 17 were female. The age range was from 0 to 93 years, with an average age of 55 years. From these clinical data, it can be speculated that the opportunistic pathogen *C. striatum* tends to cause infection and transmission in immunocompromised patients.

### 2.2. Antibiotic Resistance Profiles of Isolates

The resistance of the 54 clinical *C. striatum* isolates to nine antibiotics (penicillin, ceftriaxone, vancomycin, clindamycin, erythromycin, ciprofloxacin, linezolid, tetracycline and meropenem) was analyzed by the broth microdilution method. The drug resistance results are shown in Table 2 and Appendix A. The drug resistance to lincosamides was high, at 85.2%. The resistance to quinolones and tetracycline was 93.5% on average. The sensitivity to vancomycin and linezolid was 100%. These 54 strains of *C. striatum* were resistant to three or more types of antibiotics, so they were all multidrug-resistant.

### 2.3. Cluster Analysis of MDR C. striatum

Cluster analysis of the 54 clinical isolates of MDR *C. striatum* was carried out by built in mass spectrometry protein fingerprint software. According to the 50% similarity as the judgment level, the 54 *C. striatum* isolates can be divided into four groups (a~d) (Figure 1). The results of mass spectrometry showed that three strains in group A had high homology; these were from different samples from critical care medicine, neurology and orthopedics. Two strains in group B were from orthopedics and neurology. All the strains in group C were isolated from orthopedics, except strain Cs48. Group D contained the most strains, and these were derived from up to 13 departments, the most extensive number among the groups (Appendix A). Among them, nine strains in group C (Cs-45/50/51/52/53) and group D (Cs-46/47/49/54) were all from orthopedic secretion samples, and the collection time of the strains was concentrated in July-August 2020. Meanwhile, Cs-14/16/20/25/26/28 strains (June-August 2019) and Cs-36/37/41/42/43 strains (May-June 2019) from group D came from different departments (Figure 1). MDR *C. striatum* was found to widely exist in various departments and tended to appear in the summer (Table 1, Figure 2). Further analysis of sample types in different months showed that *C. striatum* isolates appeared almost all year round in sputum and secretion samples, while there were more isolates in summer (May to August) (Figure 2A). Similarly, by analyzing the number of department samples in different months, it was found that the sources of strain samples in summer (June to August) were rich, and mainly concentrated in orthopedics (Figure 2B).

### 2.4. Phylogenetic Evolution Analysis of MDR-C. striatum Microbiota

The *16S rRNA* gene of MDR *C. striatum* isolates was successfully cloned by molecular cloning technology. After the complete sequences were determined using BLAST (Basic Local Alignment Search Tool) in the NCBI Nucleotide Database, it was found that the *16S rRNA* gene identity of 54 clinical *C. striatum* strains was as high as 99% for *C. striatum* strains FDAARGOS 116 (CP068157.1), DSM 1566 (MN175947.1) and 190860457 (MN121138.1). MEGA 11 was used to construct the phylogenetic tree (Figure 3). It was found that the MDR *C. striatum* strains from orthopedics and nephrology had high homology, and they were located in the same branch or crossed in the same main branch. Combined with the drug resistance, it was found that *C. striatum* strains with higher homology in the same evolutionary clade had the same drug resistance (Figure 3). That is, the affinity of the strains was coupled with the drug resistance.

To further explore the possibility that there are mobile genetic elements among clinical *C. striatum* flora that drive the spread and evolution of strains, this study searched for the existence of the *integrase* gene and found that 18 strains contained this gene (Figure 4). The DNA sequence identity was 98.14% compared with that of *C. striatum* 216 (NCBI Accession: NZ_CP024932.1), and the amino acid sequence identity was 98.7% (Appendix A). Among the strains carrying the *integrase* gene were 12 strains from orthopedics, 1 strain from critical care medicine, 1 strain from neurology, 1 strain from head and neck oncology, 1 strain from otorhinolaryngology head and neck surgery and 2 strains from neurosurgery. These strains were derived from sputum (5) or secretion (13) samples (Figure 3, Appendix A).

## 3. Discussion

The latest “CHINET Surveillance Results of Bacterial Resistance in China (January–December 2021)” were released at www.chinets.com, accessed on 7 August 2022. Gram-negative bacteria accounted for 71.4%, and Gram-positive bacteria accounted for 28.6%. The emergence and spread of drug resistance in pathogens have become increasingly severe global public health problems. According to the CDC report on antibiotic resistance, more than 2.8 million antibiotic-resistant infections occur annually in the United States, resulting in 35,000 deaths. If new antimicrobial strategies are not developed, antibiotic-resistant bacterial infections will become the leading cause of death by 2050 [22], and the cumulative cost of global resistance will reach USD 100 trillion [23]. Currently, there is no public platform for reviewing death data related to multidrug-resistant infections in China. The Chinese Antimicrobial Surveillance Network (CHINET, 2018) has monitored antimicrobial resistance nationwide since 2005. These data help researchers understand the status and trend of antimicrobial resistance.

Bacteria make up about half of the body cells of the average person and are critical determinants of health and disease. Despite decades of research, the relevant and fundamental questions about how bacteria evolve and how and why multi-drug resistance and virulent strain transmission and evolution occur remain unexplained [24]. Currently, there are no guidelines for the treatment of MDR *C. striatum* infection. The optimal antimicrobial therapy is still considered controversial. In vitro susceptibility tests in this study showed that MDR *C. striatum* were sensitive to vancomycin and linezolid (consistent with Nergis Asgin et al. [25]), demonstrating their potential therapeutic value. Studies on *C. striatum* in respiratory tract infections have shown that this bacterium has strong variability and adaptability [26], and can be transmitted between patients through contact with medical staff or the hospital environment, which is prone to multi-drug resistance and can cause nosocomial transmission and outbreak [27]. This is consistent with the analysis of *C**. striatum* sample types and department distribution results in this study (Appendix A). Sputum specimens (Cs7) were first identified in critical care medicine (5 January 2019) and then successively appeared in neurology, orthopedics, nephrology and other departments. There were more sputum and secretion samples than other samples, the peak of infection was in summer (Figure 2), and the patients’ disease diagnosis was mainly associated with immunocompromised populations (Appendix A). Therefore, neurosurgery departments, orthopedics departments and intensive care units should pay attention to evaluating the clinical significance of MDR *C. striatum*.

With the genome sequencing of clinical *corynebacterium* and the study of drug resistance in recent years, the molecular mechanism of *C. striatum*’s evolution and spread is also worth further exploration. The clinical isolate *C**. striatum* M82B carries the 50 kb R-plasmid pTP10 and is resistant to chloramphenicol, erythromycin, kanamycin and tetracycline. DNA sequence analysis of the chloramphenicol-resistant region revealed the existence of a 4155 bp transposable element, Tn5564 [28]. More studies have analyzed integron and antibiotic resistance genes (ARGs) in samples collected from different locations in Antarctica [29,30,31,32]. A total of 17 microbial mats and soil metagenomes were sequenced by high-throughput sequencing, and these data were analyzed using the IntegronFinder program. Some *intI* genes were found to be similar to sequences previously determined by amplicon library analysis in soil samples collected from non-Antarctic sites. A total of 53 ARGs were found in 15 metagenomes, some encoding aminoglycoside-modifying enzymes (*AAC(6′) acetyltransferases*), some encoding class D β-lactamases (*blaOXA-205*), some encoding aminoglycoside adenyltransferase (*aadA6*), etc. [32]. In Proteobacterial organisms, a new family of integrases was discovered that mediates site-specific integration and excision of cargo gene pools in Genomic islands (GEIs). This highly chimeric cargo gene pool encodes various functions, including phage lysogeny, heavy metal resistance, metabolic enhancement and binding transfer, and many unspecified hypothetical proteins [21]. Multiple integration elements, carrying alternative integration/excision modules, have been found in several species, such as *Vibrio*, *Aeromonas*, *Salmonella*, *Pokkalibacter* and *Escherichia* [33]. The 54 strains of *C**. striatum* identified in this study, 18 of which carried site-specific integrases, are likely to be closely associated with mobile inheritance and likely participate in the process of horizontal gene transfer.

According to the Law on the Control and Prevention of Infectious Diseases, the contamination and infection of MDR *C. striatum* should be accurately determined to ensure the optimal use of antibiotics to prevent and control the aggravation of MDR *C. striatum* resistance and cause more significant loss to patients and society. The emergence of resistant bacteria has led to the post-antibiotic era. Due to the limited amount of antibiotic research and development, the difficulty of innovation, and the long cycle, new strategies are urgently needed to develop alternative treatments to prevent infection by resistant bacteria. In 2021, the “ ‘323’ Action Plan on Prominent Problems affecting People’s Health in Hubei Province (2021–2025)” proposed to put prevention first, promote a civilized and healthy lifestyle, let more people know about drug-resistant bacteria, use antibiotics correctly, establish a green and healthy life concept, and pay attention to and protect the microecological balance of the body. One of the main indicators of the ‘323’ action is the reduction of “the proportion of personal health expenditure in the total health expenditure”. In this study, 54 multidrug-resistant *C. striatum* strains from 14 clinical departments were collected and identified from the laboratory of the Affiliated Hospital of Hubei University of Arts and Science (Xiangyang Central Hospital). These strains were mainly distributed in common secretion and sputum, with there being a high incidence in summer. Phylogenetic analysis showed drug resistance and homology coupling with patients with high homology between the flora during the same period. The presence of a locus-specific mobile genetic element, *integrase* gene, in the bacterial community suggests that the resistance of this pathogenic bacteria group was dynamic and expanding. Antibiotic resistance mechanisms should be monitored during the implementation of genome research.

## 4. Materials and Methods

### 4.1. Study Design

Since 2018, *C**. striatum* has been frequently detected in the Department of Medical Laboratory at Affiliated Hospital (Xiangyang Central Hospital), bringing it to our research team’s attention. Therefore, this cross-sectional study was conducted between January 2019 and January 2022 at Xiangyang Key Laboratory of Molecular Medicine at Hubei University of Arts and Science.

### 4.2. Isolation and Identification of C. striatum Strains with MALDI-TOF MS

The study included 54 *C. striatum* strains isolated from routine clinical samples of inpatients at Xiangyang Central Hospital between January 2019 and September 2020. The strains were stored in Luria-Bertani broth with 18% glycerol at −80℃ until use. Only one strain from each patient was included. If multiple *C. striatum* isolates were recovered from a patient, only the first isolate was included in the study.

Clinical samples (sputum, blood, wound, secretion) collected from inpatients were inoculated on 5% sheep blood Columbia agar (Becton Dickinson and Company (BD), sparks, MD, USA), MacConkey (MAC) agar, and chocolate agar (BD). For the eligibility assessment of sputum specimens, WBC > 25/LP, EPC < 10/LP, and leukocyte phagocytosis of Gram-positive bacilli was conducted under the microscope. For blood culture, samples were taken from two sets of blood cultures (aerobic and anaerobic bottles) of the patient’s body (left and right arms, left and right thighs). Patient blood was inoculated into BD BACTEC Plus vials and incubated for one week in a BACTEC FX 40 (BD, MD, USA) automated blood culture system. Samples with positive signals were inoculated on 5% sheep blood Columbia agar, MAC agar, and chocolate agar plates. After an incubation period of 14–16 h, 37 °C in 5% CO_2_ atmosphere conditions, Gram staining was performed on the catalase-positive colonies and microscopically examined. When Gram-positive pleomorphic bacilli were seen, the colonies were streaked and purified, and then identified using the BD PhoenixTM automated system. There were four bottles in total, and at least two bottles of culture needed to be detected for *C*. *striatum* to be identified. If a patient from whom the blood sample came had clinical symptoms of bloodstream infection, and therefore we thought that their pathogenic bacteria were isolated, the possibility of contamination was ruled out. Identification of isolates was confirmed using the matrix-assisted laser desorption/ionization time-of-flight (MALDI-TOF) method (IVD MALDI Biotyper 2.3, Bruker Daltonik GmbH, Germany) at Molecular Microbiology Laboratory of Xiangyang Central Hospital. Strains identified as *C. striatum* by MALDI-TOF MS mass spectrometer were numbered from Cs1 to 54. Then, the bacterial homology was analyzed using the built-in protein fingerprint cluster analysis software of the mass spectrometer [34].

### 4.3. Cloning and Identification of 16s rRNA/Integrase Gene of C. striatum

The *C. striatum* isolates previously identified using mass spectrometry were further identified by *16**S rRNA* gene amplification and sequencing. High-fidelity PCR amplification was performed using the universal primers of the *16**S rRNA* gene of the *C. striatum* strain (Appendix A) [27], and no template was added as a negative control. The PCR products were electrophoresed and purified (Gel Extraction Kit, omega, Norcross, GA, USA), subcloned into the blunt-end vector (pEASY^®^-Blunt Zero Cloning Kit, TransGen, Beijing, China), transformed into *Escherichia coli* DH5α competent cells, and spread on Luria-Bertani agar (ampicillin 0.1 mg/mL). PCR was performed using the colony as template, and the positive colonies were inoculated into the broth containing ampicillin overnight (37 °C, 200 rpm). Plasmids carrying the *16**S rRNA* gene fragment were extracted with PurePlasmid Mini kit (CWBIO) and sent for sequencing (AuGCT, Wuhan, China). The amplification and identification of the site-specific recombination element, the integrase gene, was also carried out according to similar steps. Primers of the *integrase* gene (Appendix A) were designed with reference to the complete genome of the *Corynebacterium striatum* strain 216 (Accession: NZ_CP024932.1, 1618198-1619379).

### 4.4. Bioinformatics Analysis and Phylogenetic Tree Analysis of MDR C. striatum

After the successfully cloned MDR *C. striatum 16S rRNA*/*integrase* gene was sequenced, manual proofreading was performed using Chromas with reference to the positive and negative sequence maps. A homologous sequence search was performed in the GenBank database with Basic Local Alignment Search Tool (BLAST) provided by the National Center for Biotechnology (NCBI). DNA sequences were aligned with ClustalW multisequence alignment program [35]. Pairwise identity percentages between amino acid sequences were computed using the EMBOSS Needle tool (http://www.ebi.ac.uk/Tools/psa/emboss needle/ accessed date 20 August 2021). MEGA 11 program package was used to construct a phylogenetic tree with the best model (the group with the lowest BIC value: Kimura 2-parameter model and Gamma Distributed). Reliability of each tree topology was checked with 1000 bootstrap replication.

### 4.5. Antibiotic Susceptibility Testing

Susceptibility to 9 antibiotics (penicillin, ceftriaxone, vancomycin, clindamycin, erythromycin, ciprofloxacin, linezolid, tetracycline and meropenem) was tested using Microbial (Strep/Enterococcus) identification and drug susceptibility analysis system test board (Zhuhai Meihua Medical Technology Co., Ltd., Zhuhai, China) and the MicroScan WalkAway-96 Plus system (Beckman Coulter, Inc., Brea, CA, USA), as described for *Corynebacterium* [27]. The minimum inhibitory concentration (MIC) of *C. striatum* was performed strictly with the rules of CLSIM45-A3, and the antibiotic susceptibility was also determined according to this standard [36].

### 4.6. Ethical Approval

This study follows internationally recognized ethical principles, such as the Declaration of Helsinki, the international ethical guidelines for biomedical research and other relevant policies. It was approved by the Scientific Ethics Committee of Hubei University of Arts and Science (Decision no. 2022–023).

## Figures and Tables

**Figure 1 pathogens-11-01012-f001:**
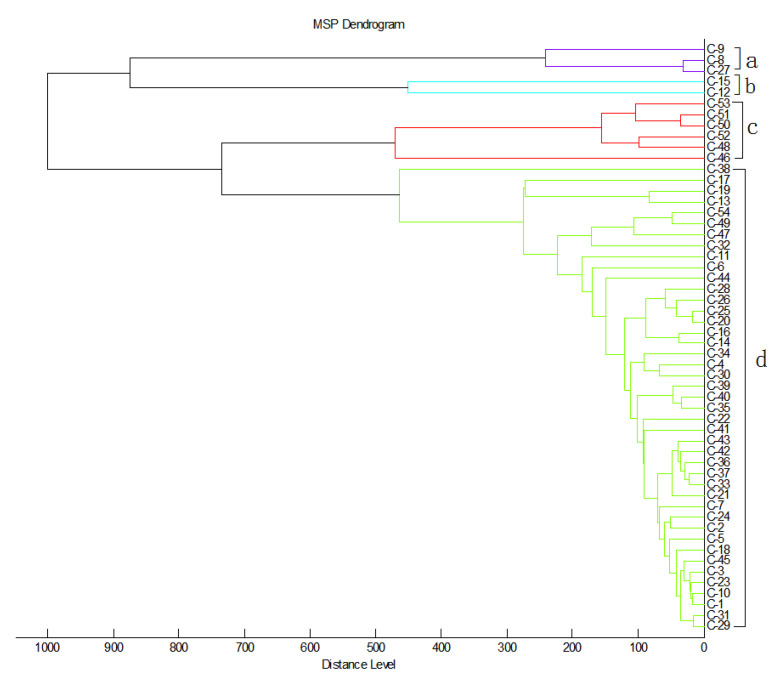
Main spectra library (MSP) dendrogram of MALDI-TOF mass spectral profiles from 54 clinical isolates of MDR C. striatum generated by the MALDI Biotyper 3.0 software. Distance is displayed in relative units.

**Figure 2 pathogens-11-01012-f002:**
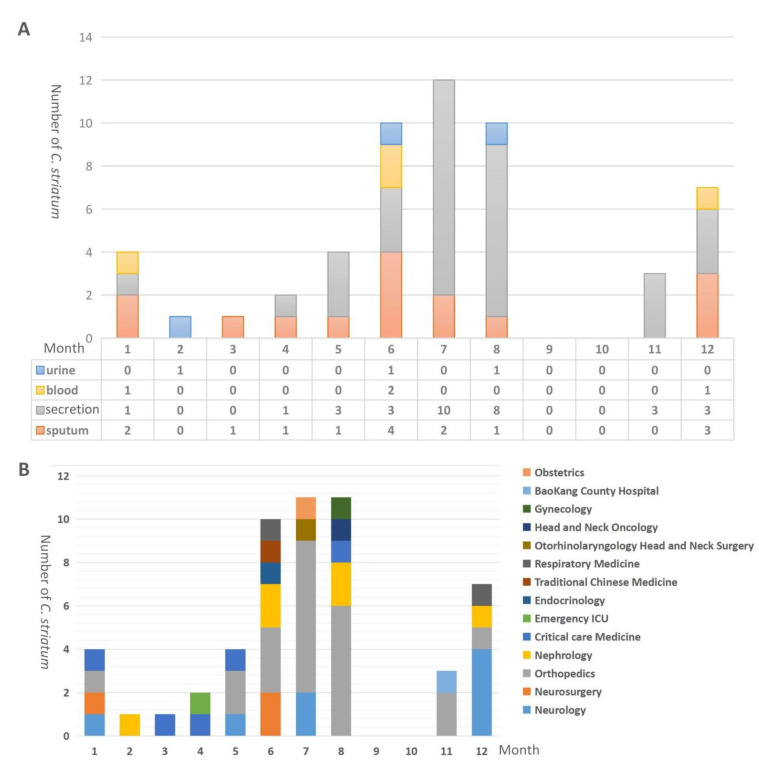
Distribution of *C. striatum* isolated from various sample types (**A**) and departments (**B**) in different months.

**Figure 3 pathogens-11-01012-f003:**
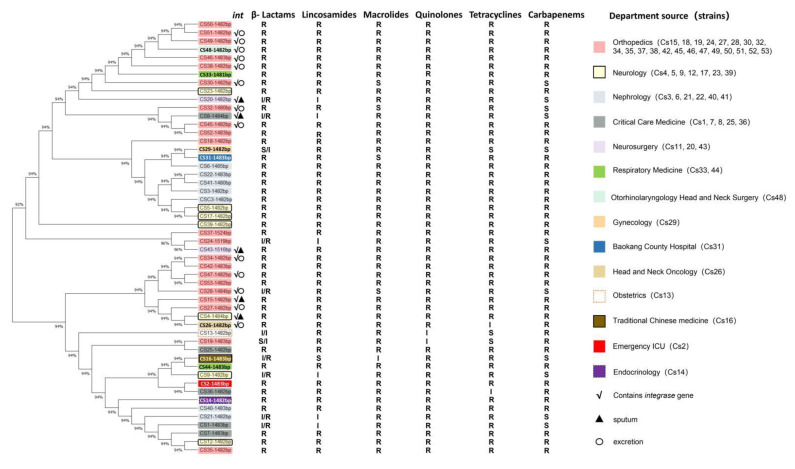
Phylogeny of MDR *C. striatum* and its origin in corresponding departments and the existence of integrase genes. The phylogenetic tree of *16S rRNA* gene sequences of *C. striatum* isolates using MEGA 11. Numbers at nodes represent percentages of bootstrap support based on a neighbor-joining analysis of 1000 resampled datasets. The details of strains’ drug resistance and *integrase* genes’ existence are marked in the middle panel (R, resistant; S, susceptible; I, intermediate). The number and color of the MDR *C. striatum* strains correspond to the source of the specimen.

**Figure 4 pathogens-11-01012-f004:**
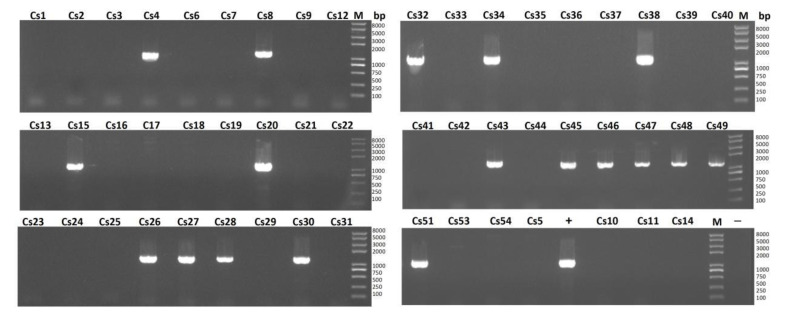
Amplification of integrase gene in 54 clinical MDR *C. striatum* isolates. Amplification of the *integrase* (*int*) gene using primer pairs (Int-1151-F/R, 1151 bp). Lane M, Trans 2K PlusII marker. +, DNA from *C. striatum* strain 216 used in PCR as positive control. −, H_2_O used in PCR as negative control.

**Table 1 pathogens-11-01012-t001:** Composition ratio of *C. striatum* samples in different departments.

Departments	Specimen Type (Number, n)	Number (Ratio %)
Orthopedics	sputum (1); secretion (21)	22 (40.7)
Neurology	sputum (6); secretion (1); blood (1)	8 (14.8)
Nephrology	urine (2); secretion (1); blood (3)	6 (11.1)
Critical Care Medicine	sputum (4); secretion (1);	5 (9.3)
Neurosurgery	sputum (3)	3 (5.6)
Respiratory medicine	sputum (1); secretion (1)	2 (3.7)
Others *	urine (1); secretion (7)	8 (14.8)
Total		54 (100)

* Otorhinolaryngology head and neck surgery (n = 1), gynecology (n = 1), county hospital (n = 1), head and neck oncology (n = 1), obstetrics (n = 1), traditional Chinese medicine (n = 1), emergency ICU (n = 1), endocrinology (n = 1).

**Table 2 pathogens-11-01012-t002:** Antibiotic-resistant species and ratios of *C. striatum* isolates.

Antibiotic Name	Drug Resistance (%)	Class of Antibiotics
Penicillin/Ceftriaxone	77.8/94.4	β-lactamases
Clindamycin	85.2	Lincosamides
Erythromycin	90.7	Macrolides
Ciprofloxacin	96.3	Quinolones
Tetracycline	90.7	Tetracyclines
Meropenem	75.9	Carbapenems

## Data Availability

Not applicable.

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
