# Peer review of "Identification and Evolutionary Relationship of Corynebacterium striatum Clinical Isolates"

_pathogens, 2022, doi:10.3390/pathogens11091012_

Round 1

Reviewer 1 Report

Dear authors, thank you for submitting to Antibiotics your work entitled "Identification and Evolutionary Relationship of Corynebacterium striatum Clinical Isolates". It focuses on a very fastidious, emerging pathogen which poses important therapeutic challenges considering the few available options. The paper is well-written, and easily readable. However I think in order to improve clarity and soundness some changes are required. Here my suggestions:

-Line 16 change sentences into "Further analysis of the..."

-Line 55: It seems some words are missing here to conclude the sentence.

-Line 73-76: Please clarify what "secretion" means, and if blood specimen are considered isolation in 1 or 2 blood sample.

-Line 79-81: Please reformulate the sentence as its meaning is not so clear. 

-LIne 82: Some more clinical data about the 54 patients are needed. Could you provide some data concerning possible immune-compromise (diabetes, cancer, hematologic disease, use of steroids)?

-Table 1: In this actual form this table is not so useful, it would be much better to introduce for each department the different sites or type of isolation. I mean in orthopedic it is extremely important to know if a bacterium is isolated from prosthesis or from swab, collection, etc. So it would be useful to discriminate for each department type of isolation to underline the possible role of Corynbacterium as pathogen or contaminant.

-Line 92: Can authors provide the MICs for the different antibiotic? It would be of extreme interest for readers.

-Line 115-116: Is it possibile to change the figure in order to introduce the departments where the samples were monthly isolated? It would give a clear idea of strain distribution and possible pattern of transmission

-Line 122, Figure 2: "Excretion" should be changed with something else is it secretion, drainage, fluid collection?

-Line 166: What about Chinese data about MDR infection-related deaths?

-Line 178: Authors should include an epidemiological analysis (timeline of appearance of strains in the different departments) in order to speculate about transmission

Reviewer 2 Report

The manuscript of Wang et al. described the isolation of C. striatum strains at the Xiangyang  Central Hospital (China) with their further characterization which demosntrated those MDR nature and clonal relationships. The study design, implementation and comunnication were good but the authors could have been emphasized their results on the MDR nature and nosocomiality importantly in the discussion. Additionally, detection of integrase genes is not informatory enough for all varieties of MGEs.

Special comments

Lines 16-17: obscure grammar, please, correct the sentence

Lines 42-43: oxytetracycline twice

Line 59: insertion sequences (!), however they 'carry' antibiotic resistance genes only in form of compound transposons

Lines 63-68: This summary is not necessary here.

Lines 87-90: This paragraph is not intelligible at al.

Table 2: Lincosamides

Lines 173-175: antibiotis are not resistant, please, correct the whole sentence

Lines 187-185: a plasmid is not resistant

Lines 187-189: refrence is missing

Lines 218-226: This part could be Conclusions with supplementing it with the study's significance
